



# Characterization of ultrafine particles and the occurrence of new particle formation events in an urban and coastal site of the Mediterranean area

Adelaide Dinoi[1], Daniel Gulli[2], Kay Weinhold[3], Ivano Ammoscato[2], Claudia R. Calidonna[2], Alfred Wiedensohler[3], Daniele Contini[1]

[1]Institute of Atmospheric Sciences and Climate, ISAC-CNR, S. P. Lecce-Monteroni km 1.2, 73100 Lecce, Italy
[2]Institute of Atmospheric Sciences and Climate, ISAC-CNR, Zona Industriale, I-88046 Lamezia Terme (CZ), Italy
[3]Leibniz Institute for Tropospheric Research, 04318, Leipzig, Germany

*Correspondence to*: Adelaide Dinoi (a.dinoi@isac.cnr.it)

**Abstract.** In this work, new particle formation events (NPF) occurred at two locations in Southern Italy, the urban background site of Lecce (ECO station) and the coastal site of Lamezia Terme (LMT station), are identified and analysed. The study aims to compare the properties of NPF events at the two sites located 225 km away from each other and

characterized by marked differences in terms of emission sources and local weather dynamics. Continuous measurements of particle number size distributions, in the size range from 10 nm to 800 nm, were performed at both sites by a Mobility Particle Size Spectrometer (MPSS). The occurrence of NPF events, observed throughout the study period that lasted five years, produced different results in terms of frequency of occurrence, 25 % of the days at ECO and 9 % at LMT. NPF events showed seasonal patterns, higher frequency during spring and summer at the urban background site, while at the coastal site

during the autumn-winter period. Some of these events happened simultaneously at both sites, indicating the occurrence of the nucleation process on a large spatial scale (regional event). Cluster analysis of 72 h back-trajectories showed that during the NPF events the two stations were influenced by similar air masses, most of which originated from the North-Western directions. Local meteorological conditions characterized by high pressure, with a prevalence of clear skies, low level of relative humidity (RH < 52 %), and moderate winds (3-4 m s$^{-1}$) dominated the NPF events at both sites. Notable differences

were observed in SO$_2$ and PM$_{2.5}$ concentrations, resulting in ~65 % and ~80 % lower at LMT compared to ECO, respectively. It is likely that the lower level of SO$_2$, recognized as one of the main gas precursors involved in the nucleation process, can be responsible for the smaller NPF frequency of occurrence (~60 % less than ECO) observed in LMT.

## 1 Introduction


The formation of new particles (NPF), by nucleation of gas-phase species and consecutive growth, is an important atmospheric process that contributes to producing high levels of ultrafine particles (UFP, diameter < 100 nm). Together with


primary emission sources, natural and anthropogenic, the nucleation represents a significant source of secondary ultrafine aerosol particles and cloud condensation nuclei (Yu et al., 2020; Merikanto et al., 2009), accounting for about 50 % of the

production of particle number concentration on a global scale. Due to their size, high number concentration, and chemical composition, these particles have profound implications on the environment, climate, and public health (Sartelet et al., 2022; Zhang et al., 2015) which emphasize their relevance.

The formation of new particles was observed and investigated in various geographic locations, under different atmospheric and environmental conditions, from which common characteristics and different peculiarities emerged (Franco et al., 2022;

Yadav et al., 2021; Jokinen et al., 2021; Baalbaki et al., 2021).

The new particle formation can occur on a local scale, typically characterized by an intense burst (strong intensity) of secondary particle formation of short duration and without subsequent growth (Dai et al., 2017), usually related to local anthropogenic emissions (Hussein et al., 2014), or else it can manifest as a part of an event which takes place on a large spatial scale (regional event). In this case, the nucleation process can originate from several hundred kilometres away from

the measurement site (Dall'Osto et al., 2013; Zhu et al., 2014; Németh and Salma 2014), and the newly-formed particles can be contemporary observed in multi-location measurements.

Such regional events were documented in the literature by various authors, i.e. in the work by Nemeth et al. (2018) 12 NPF events were observed simultaneously at three European cities located within a distance of 450 km, Budapest, Vienna, and Prague; during two years measurement campaign, Kalkavouras et al. (2020) observed 35 simultaneous NPF events that

occurred at the urban site of Athens and the regional background site of Finokalia, located 340 km away. Concomitant NPF events were also detected in the spring of 2002 in the city of Leipzig and three other locations situated 10-50 km away each other (Wehner et al., 2007), as well as regional NPF events were recorded in two sites of Finland, Hyytiälä and Värriö, that situated within a distance of 220 km (Hussein et al., 2009). It was noted that most of these events occurred in analogous conditions, under the influence of air masses with the same origin and enriched with sulphur-rich precursors (Kalkavouras et

al., 2020), and exhibiting similar dynamic characteristics. NPF events were usually detected in ground-based measurement stations, but a number of observations were also performed at high altitudes, on mountain tops (Bianchi et al., 2016; Rose et al., 2015), by balloons and through aircraft (Mirme et al., 2010; Schroder and Strom, 1997) leading to hypothesize that the nucleation process can be triggered within the atmospheric column, at different heights (Boulon et al., 2011; Minguillón et al., 2015; Querol et al., 2017), both throughout the boundary layer and in the free troposphere. At upper atmospheric levels,

nucleation would be facilitated by the higher dilution of the pollutants (low condensation sink) and by enhanced photochemical conditions. Once nucleated, the particles can be transported over long distances by the circulation of air masses. Where and at which altitude the mechanism is triggered is still unclear (Boulon et al., 2011), but in this process atmospheric dynamics could play a crucial role in the mixing of precursor gases and/or pre-existing particles, influencing their occurrence and spatial distribution (Wehner et al., 2015).

To date, the variability of the conditions and the complexity of the phenomenon make our knowledge of this process still fragmentary in many aspects. While it is well known that solar intensity, low relative humidity, and the availability of high



levels of aerosol nucleation precursors (sulphur dioxide, ammonia, amines, VOCs) are common features in the NPF (Hao Wu et al., 2021), the chemical/physical mechanisms involved in the process still remain uncertain and further insights are required.

In this paper, we summarize the results of a long-term study on the new particle formation events carried out at two sites in southern Italy, Lecce (ECO station) and Lamezia Terme (LMT station). To the best of our knowledge, this work represents one of the few long-term studies on NPF events conducted in the Mediterranean area and the first one conducted in Italy. Based on a 5-yr period, simultaneous measurements of aerosol particle number size distribution were analysed with the aim to characterize, investigate and compare the relationship of NPF events with meteorological and pollution data at a coastal

and at an urban background site. To support interpretations, simultaneous measurements of $SO_2$ and $PM_{2.5}$ ambient concentrations were carried out at both ground monitoring stations. The role of air mass and local meteorological factors were also investigated.

## 2 Methodology

### 2.1 Study area

Measurements of particle number size distribution (PNSD) were made from January 2015 to December 2019 at two sites of southern Italy, Lamezia Terme Observatory (LMT, 38.88° N, 16.23° E, 6 m a.m.s.l.,) in Calabria region, and Lecce Observatory (ECO, 40.20° N, 18.07° E, 50 m a.m.s.l.) in Puglia region. The locations of the measuring sites are depicted in Fig. 1. ECO and LMT, both regional stations of GAW/ACTRIS networks, are representative of urban backgrounds and

suburban/coastal sites, respectively. ECO observatory is located inside the University Campus (Dinoi et al., 2020), at about 5 km southwest of the municipality of Lecce, while LMT observatory is located about 17 km from the urban city and about 600 m inland from the Tyrrhenian coastline, at an elevation of 6 m above sea level (a.s.l.). The two sites are about 225 km as the crow flies. The municipalities cover areas of 238 km² (Lecce) and 160 km² (Lamezia) with population of 95,000 and 70,452 inhabitants respectively. Their climate is Mediterranean, albeit characterized by different meteorological dynamics.

Different types of emission sources and levels of pollutants (more details from Dinoi et al., 2021 b; Calidonna et al., 2020; Cristofanelli et al., 2018) affect the two sites. The main aerosol sources in the ECO observatory are the emissions of vehicular traffic and biomass combustions, which are added to natural and anthropogenic long-range contributions (Donateo et al., 2018; Conte et al., 2020). The site is characterized by frequent conditions of clear skies during the whole year, with mild autumn-winter seasons and warm spring-summer. Due to its location, away from the urban agglomeration, the LMT

observatory is weakly affected by the main emission sources deriving from anthropic activities. Being on the coast, local weather is influenced by a system of land-see breezes that guarantees a mild and temperate climate together with an effective dilution of air pollutants.

### 2.2 Instruments




Aerosol PNSD from 10 to 800 nm, with a time resolution of 5 minutes, were collected using a TROPOS-type custom-built MPSS (Wiedensohler et al., 2012), designed and manufactured according to EUSAAR/ACTRIS recommendations. The instruments used are the same at the two stations as well as the calibration and data quality control procedures used. The two MPSS consist of a bipolar diffusion charger (Ni-63), a differential mobility analyzer (Vienna-type DMA, length 28 cm), and

a condensation particle counter (CPC, model: TSI 3772, TSI Inc., Rome, Italy).

It is a closed-loop system, with a 5:1 ratio between the sheath and aerosol flow, where the sample air is drawn into the instrument at a flow rate of 1 l/min, while the sample humidity is regulated below 40 % by a Nafion dryer. The quality of measurements of both instruments was routinely checked and all data were corrected for particle losses by diffusion and negatively charged particles according to the recommendations of Wiedensohler et al., (2012).

Concentrations of sulphur dioxide ($SO_2$) were measured using Thermo Instruments analyzers, TEI 43i, and $PM_{2.5}$ mass concentrations using low volume samplers with a β-ray attenuation method (SWAM 5a Dual Channel Monitor (FAI Instruments). Local meteorological parameters, air temperature (T), relative humidity (RH), precipitation (Rain), wind speed (WS), and wind direction (WD) were monitored by automatic weather stations (Vaisala WXT 520), located in the two observatories while solar radiation (solar flux) was collected by CNR 4 net radiometers (Kipp&Zonen). Air mass back

trajectories were calculated using the HYSPLIT 4 model developed by NOAA/ARL, a single particle Lagrangian trajectory dispersion model (Stohl 1998).

**2.3 Evaluation of NPF events**

Classification of NPF events was performed by visual inspection of daily contour plots (Dal Maso et al., 2005). Examining

the time evolution of the particle number size distribution, three main classes were detected: NPF events, non-event, and undefined events. NPF events contain cases where a significant increase in the number concentrations of ultrafine particles and growth toward larger diameters was observed for at least 3-4 h continuously, displaying the shape of a "banana". "Non-events" are the days without new particle formation, while "undefined events" group ambiguous cases, with unclear formation and growth, or the occurrence of newly formed particles below 20 nm without the next phase of growth.

Particle growth rate (GR) was calculated from time evolution of the mean geometric diameter $D_p$ inside the sub-50 nm mode particles using Eq. (1) (Kulmala et al., 2012):

$$GR(nm\ h^{-1}) = (D_{p2} - D_{p1})/(t_2 - t_1), \tag{1}$$

with $D_{p1}$ and $D_{p2}$ the geometric diameter at the time of start $t_1$ and end $t_2$ of the growth event.

**3 Results and discussions**

**3.1 Classification of NPF events**



The relative frequency of the different classes, events, non-events and undefined events, was calculated for both sites and summarized in Table 1.

The available measurement days were 1423 at ECO and 1440 at LMT with a data coverage of ~78 %. The missing days were due to technical/maintenance problems of the MPSS or measurement failures. Of these, 25 % and 9 % of days were identified as NPF events while 4% and 3% as undefined events at ECO and LMT, respectively.

We found that the percentage frequency of NPF events is higher in ECO than in LMT, with differences in the monthly and seasonal occurrence of the events (Fig. 2).

At the ECO site the highest frequencies of NPF events were observed in March and September (~30 %) and the lowest in November and December (12 %-16 %), confirming what was already observed in Dinoi et al. (2021a). At LMT, March (19 %) and October (16 %) showed the highest frequencies while July and August were the lowest ones (2-3 %). Consequently, this is reflected in a different seasonality of events, more frequent in spring and summer in ECO, and in autumn and winter in LMT. Of all events detected during the study period, 50 were observed simultaneously at both sites. Due to the complementary frequency, most of them occurred in the colder months, ~ 23 % in winter and spring, 14 % in summer and 39 % in autumn.

The annual frequency of NPF is in good agreement with frequencies (10 %-36 %) found in other studies carried out in the Mediterranean area (Baalbaki et al., 2021). The seasonal variability observed in the ECO site was similar to the results found in several other locations (in Europe and around the world) and can be ascribed to the higher emissions of biogenic aerosol precursor compounds and photochemical processes promoted by the higher temperature during the warmer months (Liao et al., 2014; Asmi et al., 2016). However, the opposite trend displayed by the LMT site is not a novelty since it has also been observed, albeit less frequently, in other southern Europe sites. In Spain and Greece, for example, Bousiotis et al. (2021) found the occurrence of a larger number of NPF events just during winter and that might be linked to the specific meteorological conditions of the study area.

The temporal evolution of the events was very similar between the two sites, the start times were in the morning between 9:00 – 12:00 with a median duration of 5-6 hours, similar to what was observed by Dinoi et al. (2021a).

Typically, the particle growth rates were high during the first few hours of nucleation and then decreased to a few nanometers per hour within 3-4 hours after nucleation. The growth rate of particles varied between 3 nm h$^{-1}$ and 14 nm h$^{-1}$ with a mean and standard deviation of $7.5 \pm 3.3$ nm h$^{-1}$ in ECO, and between 2.5 nm h$^{-1}$ and 10 nm h$^{-1}$ with a mean and standard deviation of $6.1 \pm 2.3$ nm h$^{-1}$ in LMT. In general, lower GR is observed in a clean environment and vice versa, but because many factors are involved in the NPF process, this behavior is not always observed.

## 3.2 Atmospheric particle number concentration (PNC)

A statistical overview of the emission levels and contributions of three diameter modes particles, nucleation ($N_{nuc}=N_{10-20}$), Aitken ($N_{Aitk}=N_{20-100}$), and accumulation ($N_{acc}=N_{100-800}$), to total particle number concentration ($N_{tot}=N_{10-800}$) was made. For





165   each observatory, arithmetic means ± 1 standard deviation, medians, and percentiles (25th-75th) of PNC were calculated

from the daily data and summarized in Table 2. The mean value of total PNC was 4408 ± 2240 cm⁻³ at LMT and 7783 ±

3381 cm⁻³ at ECO. Although significantly higher concentrations (about 40 %) were found at the urban background site

compared to the coastal site, the contribution of each particle fraction to total particle number concentration was very similar,

24 %, 49 %, and 27 % in LMT and 20 %, 52 % and 28 % in ECO, for nucleation, Aitken, and accumulation mode particles,

respectively. In both cases, Aitken mode particles represented the largest fraction of particle number concentrations.

The levels measured at two sites are within the range reported for similar locations observed in other European sites

(Casquero-Vera et al., 2020; Asmi et al., 2011; Putaud et al., 2010; Kalivitis et al., 2019; Kalkavouras et al., 2020). The

monthly variability of PNC, averaged over the whole study period, is shown in Fig. 3.

In ECO and LMT the highest monthly average concentrations of Aitken mode particles were observed during winter (~5650

cm⁻³ and ~2680 cm⁻³), approximately 45 % in ECO and 35 % in LMT higher than in summer (~3200 cm⁻³ and ~1700 cm⁻³).

In ECO accumulation mode particles ranged from~3100 cm⁻³ in winter to ~1750 cm⁻³ in spring, while in LMT from~1370

cm⁻³ in summer to ~980 cm⁻³ in autumn. The increase in $N_{acc}$ concentration during the summer can be ascribed to the dry

conditions that characterize the Mediterranean summers (Pikridas et al., 2018). Nucleation mode particles displayed a similar

seasonal pattern to Aitken mode particles in LMT (from ~800 cm⁻³ in summer to~1450 cm⁻³ in winter), while in ECO values

slightly higher were observed during spring/summer, ~1900 cm⁻³, than in winter ~1350 cm⁻³. The higher concentrations

recorded during cold months are generally associated with enhanced anthropogenic emissions from fossil fuel combustion

and biomass burning, due to residential heating, and unfavourable meteorological conditions for pollution dispersion, such as

frequent occurrences of stagnant weather and temperature inversion.

A clear diurnal pattern in each mode particle number concentration was observed in every season. Fig. 4 shows the trend of

each mode fraction considering separately the days of NPF events (E, solid line) and the days of non-events (NE, dashed

line). Nucleation, Aitken, and accumulation mode particles have very similar behaviour during non-events and, except for

the different concentrations, both sites show a pronounced diurnal cycle, mostly in cold seasons, with a morning and evening

peak. The two peaks, shifted by one hour between spring-summer and autumn-winter (due to the daylight savings time), are

likely due to vehicular emissions, most intense during the morning and evening rush-hour. The diurnal variation of

concentrations at the ground level is governed not only by the emission sources but also by the dynamics (height and

stability) of the PBL. Stagnant conditions and low PBL during early mornings and late evening exacerbate morning/evening

rush-hour pollution and help to build up pollution levels associated with primary emissions, mainly from traffic (Backman et

al., 2012). At noon, the height of the PBL increases favouring the mixing of pollutants throughout the PBL and limiting their

accumulation at ground level. A different situation was observed for nucleation mode particles, wherein both sites, together

with the two peaks of rush hours, further picks are present around noon, most marked in summer, spring, and winter in ECO,

and spring, summer, and autumn in LMT. Less pronounced are instead those in winter and autumn in LMT and ECO

respectively. The difference in number concentration attributed to nucleation processes is notable by which the

concentrations of particles in nucleation mode are strongly influenced. In Table 3 reported mean values ± 1 standard





deviation of nucleation, Aitken and accumulation of PNC (cm$^{-3}$) related to event days (E) and non-event days (NE) for each

season (winter W, spring Sp, summer S, and autumn A) in both sites. Nucleation mode particles show an increase of 52 %, 65 %, 61 %, and 49 % in winter, spring, summer, and autumn in LMT, and of 47 %, 52 %, 55 %, and 39 % in ECO. The contribution of NPF to the number concentration is also observed in the Aitken mode particles more noticeable in the LMT site with 30 % in autumn-winter and 41 % in spring-summer, and with 21 % only in spring-summer in the ECO site. No contribution is observed in the concentration of accumulation mode particles.

Similar observations have been reported in Dinoi et al. (2020, 2021a), Kalivitis et al. (2019), Cusack et al. (2013) for the western Mediterranean sites where the diurnal variation in nucleation mode particles presents a clear maximum at noon under both polluted and clean conditions.

The relative increase in particle number concentration due to the NPF process was also quantified with the nucleation strength factor (NFS, Salma et al., 2014) following Eq. (2):


$$\text{NFS} = \frac{(N10{-}100)/(N100{-}800)E}{(N10{-}100)/(N100{-}800)NE} \tag{2}$$

considering the N10−100 / N100−800 concentration ratios for nucleation days at the numerator and the N10−100 / N100−800 concentration ratios for not nucleation days at denominator. Depending on the value, NFS <1.0, 1.0 <NFS <2.0,

or NFS <2.0, the nucleation process can be considered negligible, comparable to the other sources or the main contributor. In LMT the mean values of NFS were 2.0, 2.1, 2.3, and 1.9 while in ECO 1.7, 1.6, 1.9, and 1.9 in winter, spring, summer, and autumn, respectively. These results point out that the particles produced by the NPF process can become the dominant sources in a clean environment compared to more polluted areas.

**3.3 Factors associated with NPF events**

Since there are many factors involved in the process, direct measurements of PM$_{2.5}$ and SO$_2$ concentrations, and meteorological variables, performed at both sites, including the condensation sink were compared. The CS was estimated from measurements of the particle number size distribution between 10 and 800 nm, more details on the CS calculation are in the literature (Dal Maso et al., 2005; Kulmala et al., 2004 and references therein). There were visible differences in

pollution load (Fig. 5) at the two sites. The average PM$_{2.5}$ and SO$_2$ concentrations during event days (columns) were 2.5±1.5 µg m$^{-3}$ (ranging from 2.0 to 3.8 µg m$^{-3}$) and 0.4±0.3 ppb (from 0.2 to 0.6 ppb) for LMT, while they were 12.0±6.0 µg m$^{-3}$ (from 9.5 to 15.3 µg m$^{-3}$) and 1.1±0.4 ppb (from 0.8 to 1.3 ppb) for ECO. These values were compared with mean values of PM$_{2.5}$ and SO$_2$ measured on non-event days (dashed lines), 6.1±2.5 µg m$^{-3}$, 0.2±0.2 ppb in LMT, and 15.5±8.0 µg m$^{-3}$, 0.9±0.4 ppb in ECO. In both sites, statistically significant differences (p-value <0.05 level, by Wilcoxon-Mann-Whitney test)

emerged between events and not-events for PM$_{2.5}$ and SO$_2$, highlighting the different role they may have played in the new particle formation. Regarding CS, it was (0.7 ± 0.3) x 10$^{-2}$ s$^{-1}$ (0.005 - 0.009 s$^{-1}$) for LMT and (1.0 ± 0.6) x 10$^{-2}$ s$^{-1}$ (0.008 -



0.01 s⁻¹) for ECO, within the range of coastal and urban environments (Kalivitis et al., 2019; Baalbaki et al., 2021; Salma et al., 2016). In this case, comparison with values measured during non-event days, $(0.7 \pm 0.4) \times 10^{-2}$ s⁻¹ in LMT and $(1.3 \pm 0.5) \times 10^{-2}$ s⁻¹ in ECO showed no noteworthy differences. The two sites showed evident differences in both concentrations, with

$SO_2$ ~65 %, $PM_{2.5}$ ~80 %, and CS ~30 % lower in LMT than ECO. This underline how they are affected in different ways by anthropogenic contributions, and the lower levels of $SO_2$ could be responsible for the reduced NPF frequency (~65 % less than ECO) observed in LMT. In general, particulate matter concentration has a direct effect on the condensation sink that quantifies how rapidly a condensable gaseous compound condenses on available aerosol particles (Kerminen et al., 2018). As known, new particle formation is "discouraged" by the high concentration of pre-existing particles because they increase

condensation sink (CS) for vapours. Therefore, it was surprising to discover the low frequency of nucleation events occurred at LMT that being a cleaner site compared to Lecce we would have expected the opposite result (Dinoi et al., 2017; Dinoi et al., 2021b). Lower CS would favour nucleation proportionally to the amount of condensable vapours available (Saha et al., 2018). In this case, LMT exhibits low levels of both factors, $SO_2$ and CS (so as $PM_{2.5}$) which would explain the observed results. The proportionality between the lower frequency of nucleation events and the lower $SO_2$ concentrations observed

between the two sites was already noted in other works. Saha et al. (2018) highlight how, in the urban background of Pittsburgh, the large reduction of $SO_2$ concentrations (~90 %) over 15 years resulted in the reduction of nucleation frequency of ~70 %, as well as Kyrö et al. (2013) observed that the decrease in new particle formation days is the result of the reduction of sulphur emissions originating from Kola Peninsula (Russia). Hamed et al. (2010) observed at the atmospheric research station of Melpitz (Germany) that decreasing $SO_2$ concentrations by an average of 65 % led to a 45 % decrease in

the frequency of NPF events, while Gaydos et al. (2005) simulated that a substantial reduction in $SO_2$ (more than 40 %) would reduce proportionally nucleation with different effects depending on the season.

**3.4 Meteorological conditions associated with NPF events**

The comparison of the main meteorological parameters was carried out to better characterize the occurred NPF also in terms of the local climate of the two study areas. The monthly average values of solar flux, relative humidity, and wind speed were

calculated starting from the daily averages. All NPF events happened in comparable weather conditions featured by high pressure, with a prevalence of clear skies, typical of the Mediterranean climate. First of all, we considered solar flux, one of the main factors involved in the nucleation process (Fig. 6a). Monthly mean values showed an intense solar flux both in LMT (from 260 to 512 W m⁻²) and in ECO (from 230 to 510 W m⁻²) during events (columns). These values were very similar to those measured during non-event periods (dashed) except at the LMT site where higher mean values are observed

in the colder months, differences which however are not statistically significant. During event days, the RH was comparable at the two sites: $52 \pm 4$ % in LMT and $50 \pm 7$ % in ECO, and it was about 25 % lower than that on non-event days (Fig. 6b). This result is not a novelty because lower RH is usually observed during NPF events in both clean and polluted environments (Kerminen et al., 2018). On average, monthly wind speeds were higher during events in both sites, $4 \pm 1$ m s⁻¹



for LMT (from 2.9 to 5.4 m s$^{-1}$) and 3 ± 1 m s$^{-1}$ for ECO (from 2.6 to 4.3 m s$^{-1}$), respectively about 10% and 40% higher than
the wind speed observed on non-event days (Fig. 6c). The difference is statistically significant only for ECO (p-value <0.05).
Regarding the wind directions (Fig. 7), an interesting difference was found between the two sites. In ECO the dominating
wind direction was mainly from N–NNW, with a frequency of occurrence of 75 %. In LMT observational data revealed two
predominant wind directions, from E-NE and W-NW, with a frequency of occurrence of 30 % and 55 %, respectively,
corresponding to a land-sea breeze system.

The two prevalent wind directions together with the higher values of wind speed measured in LMT (both on events and on
non-event days) are typical of a coastal area where generally, thanks to the circulation of the sea-land breeze, wind speed is
approximately 20 % greater than in the innermost (inland) sites and almost constant during the whole year (Jensen et al.,
2017). This process, able to influence the diffusion of atmospheric pollutants and their dilution, could explain the low and
almost constant (lack of seasonality) levels of PM measured at LMT.

## 3.5 The role of air masses associated with NPF events

As well known, the origin and pathways of air masses can contribute significantly to the occurrence of NPF processes
(Wonaschutz et al., 2015). Depending on their advection pattern, air masses can be affected by emission sources and
chemical/physical processes able to influence the properties of the pollutants during transportation. With this aim, the role of
air masses was investigated through the analysis of the back trajectories of all days identified as NPF events. 72 h back
trajectories were computed each day at 08:00 UTC, with 6 h resolution, at the 500 m height arrival above ground level
(AGL), using the HYSPLIT 4, single-particle lagrangian trajectory dispersion Model, developed by NOAA/ARL (Draxler
and Hess, 1998). We considered only the back trajectories at an altitude of 500 m because they can be better correlated with
ground-based measurements we made and, therefore, more representative of the atmospheric boundary layer in which air
pollutants are well mixed by horizontal and vertical advection.

The analysis of the back trajectories allows us to gain qualitative information on the paths of the air masses associated with
NPF events, and although they can be affected by an error estimated between 15 % and 30 % (Stohl, 1998; Draxler and
Hess, 2004), for the purposes of this paper we considered negligible such inaccuracy. The main transport pathways to the
study sites were identified by cluster analysis of back trajectories. Fig. 8 shows 6 centroids, each representative of a group of
trajectories, labelled according to the direction of the cluster, frequency of occurrence, and length. Short trajectories are
indicative of slow-moving air masses while long trajectories are for fast flows; different lengths can have implications on a
load of pollutants transported (associated) by air mass.

As represented in Fig. 8, during the occurrence of NPF events the two stations were influenced by similar air masses, most of
which were enclosed in the north-western quadrant.

Air masses arriving at LMT and originating from the North-Northwester sector exhibited an occurrence rate of 38 % (17 %
long and 20.5 % medium trajectories), followed by 62 % of air masses from the West-Northwest sector (9.8 % long, 19.6 %





medium and 33 % short trajectories). At ECO, air masses originating from the North-Northwester sector showed a percentage of 20 % (medium trajectories), those from the West-Northwest sector an occurrence of 55 % (21.6 % long and 33.4 % short trajectories), while another important contribution, 25 % (short trajectory), was associated to Eastern sector. All the air masses flew over both the European continental areas highly anthropized and the marine environment before reaching
the receptor points.

We also focused on "common NPF events" considering only the air masses pathway related to these days detected during the study period. From 3-days back-trajectory analysis, emerged that the back-trajectories of common events exhibit similar characteristics in terms of origin and pathways to what already observed, but in addition two different cases were detected, in the first the trajectory passes from ECO before reaching LMT, while in the other case, vice versa the air mass passes from
LMT and then reaches ECO. Out of 50 common NPF events, 31 were in the first case and 19 in the second. Two representative examples of back-trajectories for each case are depicted in Fig. 9 and show that there is not a preferential path that can characterize them, because in both cases we find trajectories that come from both Eastern and Western Europe. These events occurred almost synchronously at the two sites, with a difference in starting time not greater than 30 minutes.

The factors that characterized the concomitant events of NPF (Table 4) are similar to those observed for the non-concomitant
events, with concentrations of $PM_{2.5}$ and $SO_2$ and CS in LMT lower than ECO, and similar meteorological conditions, low RH values (40-56 %), moderate wind speed ~ 4 m s$^{-1}$, and good solar irradiation, from 190 to 350 W m$^{-2}$ in ECO and from 310 to 500 W m$^{-2}$ in LMT.

The simultaneous observation of these events indicates that the formation of new particles has a wide horizontal extension and can be seen as a large-scale phenomenon. It is probably that the air masses already contain particles which have been
formed by nucleation somewhere and then transported. Or during their travel the air masses are enriched with gaseous precursors deriving from anthropogenic emissions and/or from biogenic production such as to foster (potentially) NPF processes, even in locations far from the sources. Sulphuric acid, for example, originated in the atmosphere from the oxidation of sulphur dioxide and is a key compound involved in atmospheric nucleation as it is MSA, generated from biological marine processes. Also, other precursors are important in the nucleation process, and the presence of additional
chemical species such as ammonia, amines, or organic species can stabilize the nucleation clusters, decrease evaporation and, therefore, improve the nucleation of sulfuric acid particles (Almeida et al., 2013).

It is interesting to note the lack of air masses associated with NPF events coming from the southern sector. These air masses, because of their pathway, are usually characterized by high levels of dust and humidity collected during the crossing of the Mediterranean Sea. High levels of dust can counter the NPF process because can suppress photochemical activity by
scavenging reactive gases and condensable vapors (De Reus et al., 2000; Ndour et al., 2009). These results emphasize the role that air masses have not only in terms of transport of precursors but also in synoptic atmospheric conditions to them associated.

**4 Conclusions**





New particle formation events were studied and compared at two sites in southern Italy, ECO (Lecce) and LMT (Lamezia
Terme) observatories, over a period of five years. The nucleation events occurred with a different frequency, 25 %, and 9 %,
and seasonality, highest in spring-summer and autumn-winter at ECO and LMT, respectively.

Throughout the investigation period, 50 simultaneous NPF events (14 % of the cases in ECO and 40 % in LMT) were
identified at both sites, many of which occurred in the colder months, indicating that the NPF process was affecting a large
spatial extent.

The NPF days were characterized by lower $PM_{2.5}$ concentrations (~60 % and 22 %) and higher $SO_2$ concentrations (~50 %
and 20 %) compared to non-event days in LMT and ECO respectively, while the condensation sink, calculated from 10 to
800 nm, seems not to be significantly different during events and non-event days.  Marked differences in $PM_{2.5}$, $SO_2$, and CS
levels were observed between the two sites, indicating the minor anthropogenic influence to which LMT is subjected.

Common meteorological features were observed during NPF events occurred in conditions of high pressure, low RH (~52
%), and moderate wind speed (3-4 m s$^{-1}$). The study of the back trajectories associated with the events also highlighted a
common origin of the air masses, both of continental and marine origin, from the North-Northwest directions, suggesting
that the chemical compounds involved in the NPF could have been transported by the air masses. Our founding let us assume
that the lower levels of $SO_2$ (-60 % with respect to ECO) together with a different chemical composition of the aerosols and
different local meteorology might be the reason for the lower frequency of events occurring in LMT, -60% with respect to
ECO. The results presented in this paper are a contribution toward a better understanding of the complex NPF phenomenon
in central Mediterranean area which will require further investigations and measurements of different precursors (such as
ammonia, amines, VOCs) involved in the process that were not considered in this work.

**Data Availability:** Data is available upon request.

**Author contributions**: AD and DC conceptualized the study design. AD, DG and KW collaborated to data collection and
post-processing. AD and DC prepared the draft.  AD, DG, KW, IA, CC, AW, and DC, collaborated to interpretation of
results, wrote, read, commented, and approved the final manuscript.

**Competing interests:** The authors declare no competing interests.

**Acknowledgements:** This work was supported by the project CIR01_00015 - PER-ACTRIS-IT "Potenziamento della
componente italiana della Infrastruttura di Ricerca Aerosol, Clouds and Trace Gases Research Infrastructure - Rafforzamento
del capitale umano" - Avviso MUR D.D. n. 2595 del 24.12.2019 Piano Stralcio "Ricerca e Innovazione 2015-2017".

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



**Figures**

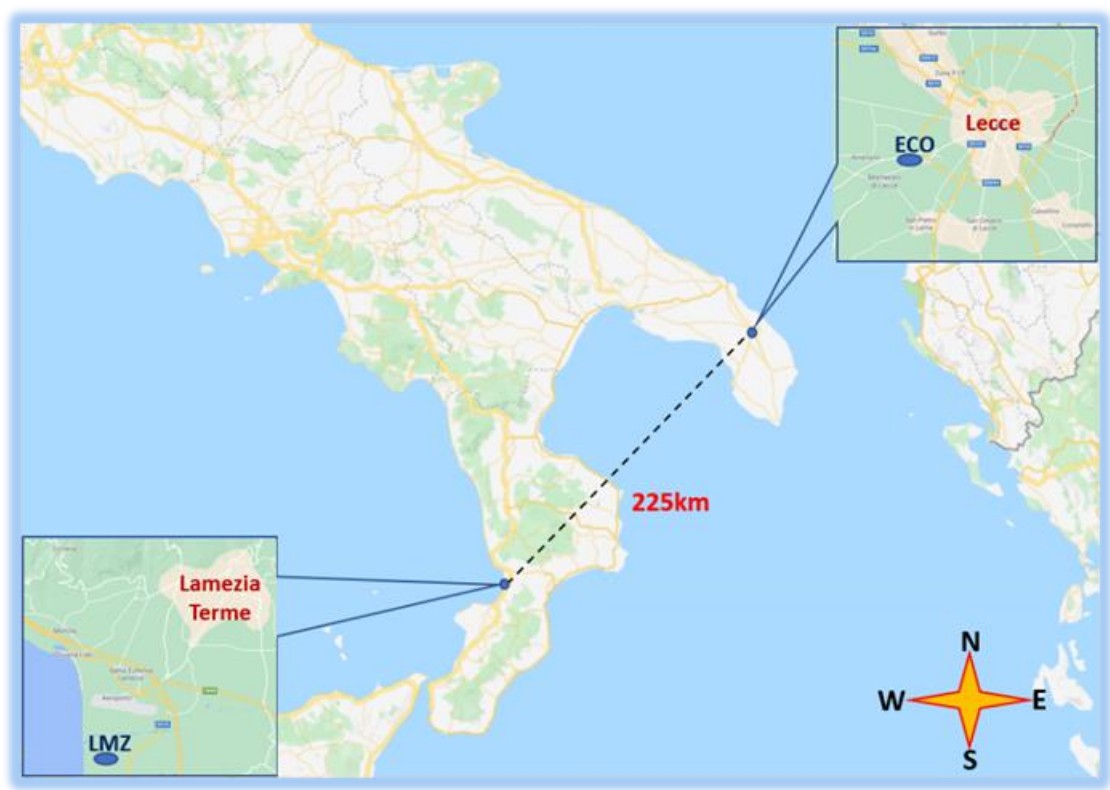

**Figure 1:** **Map of the exanimated area with the locations of the two environmental-climate observatories, ECO and LMZ. The map was retrieved from © Google Maps (Map data © 2022).**

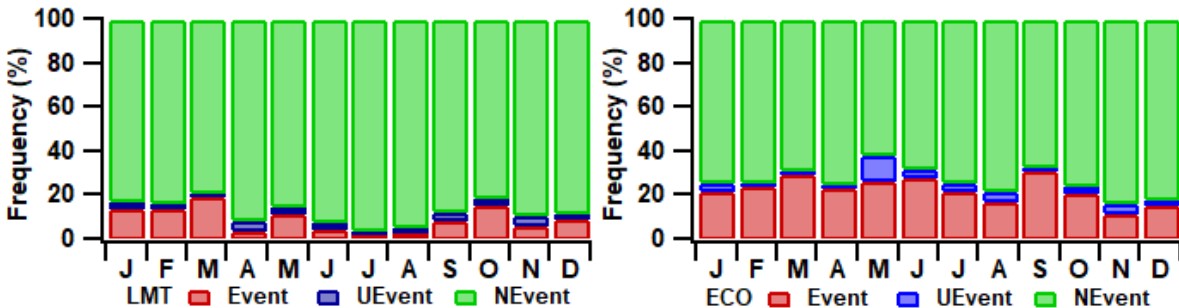

**Fig. 2. Monthly percentage of occurrence of Events (red bars), Undefined events (blue bars), and Non-Events (green bars) related to available measurement days at LMT and ECO.**




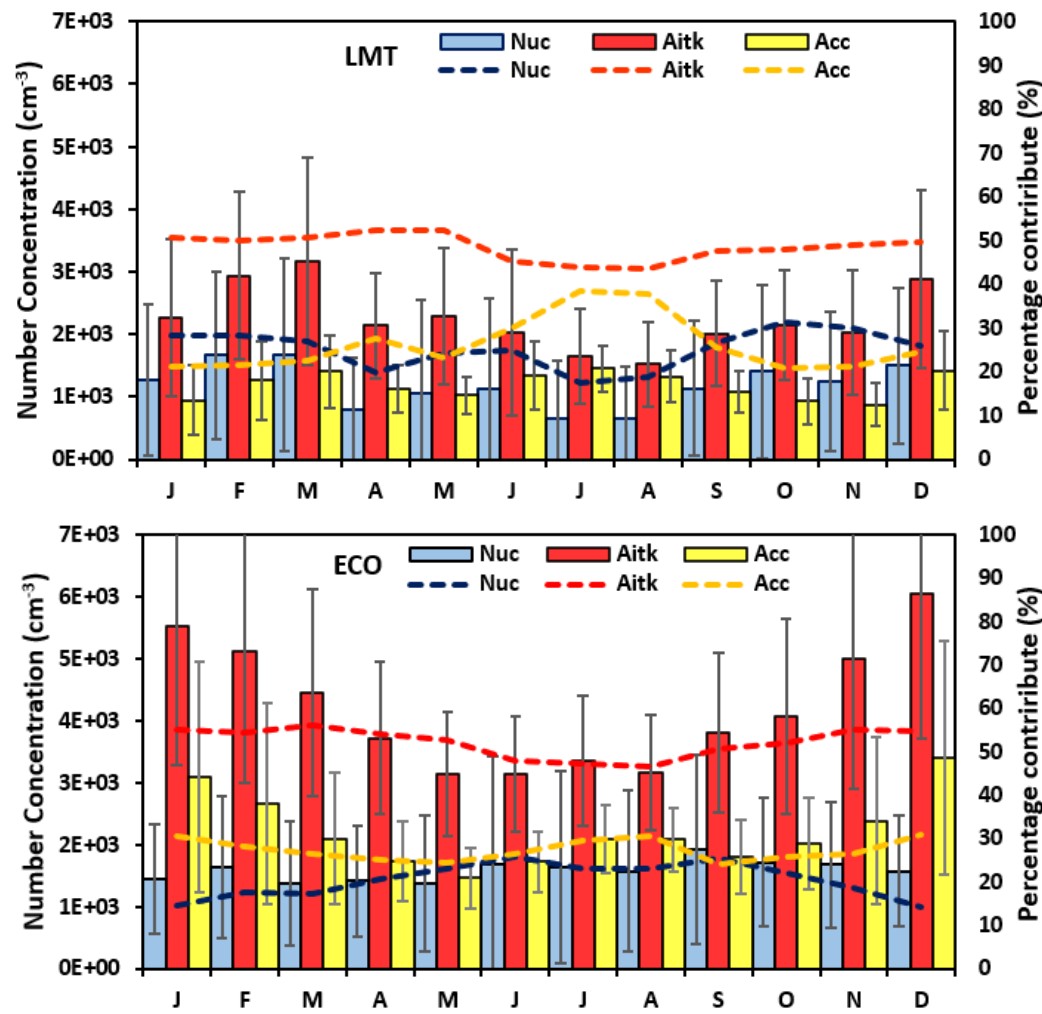

Fig. 3. Average monthly variation (bars) of the nucleation, Aitken, and accumulation mode particle number concentration over the whole period of study at LMT and ECO. The dashed lines represent the monthly percentage contribution of each particle fraction to total particle number concentration.






**Fig. 4. Average diurnal variation in nucleation, Aitken, and accumulation mode particle number concentration (from top to bottom) over the whole period of study at LMT and ECO. Solid lines are for NPF events days (E) and dashed lines for non-events days of (NE), red for winter, grey for spring, blue for summer, and yellow for autumn.**





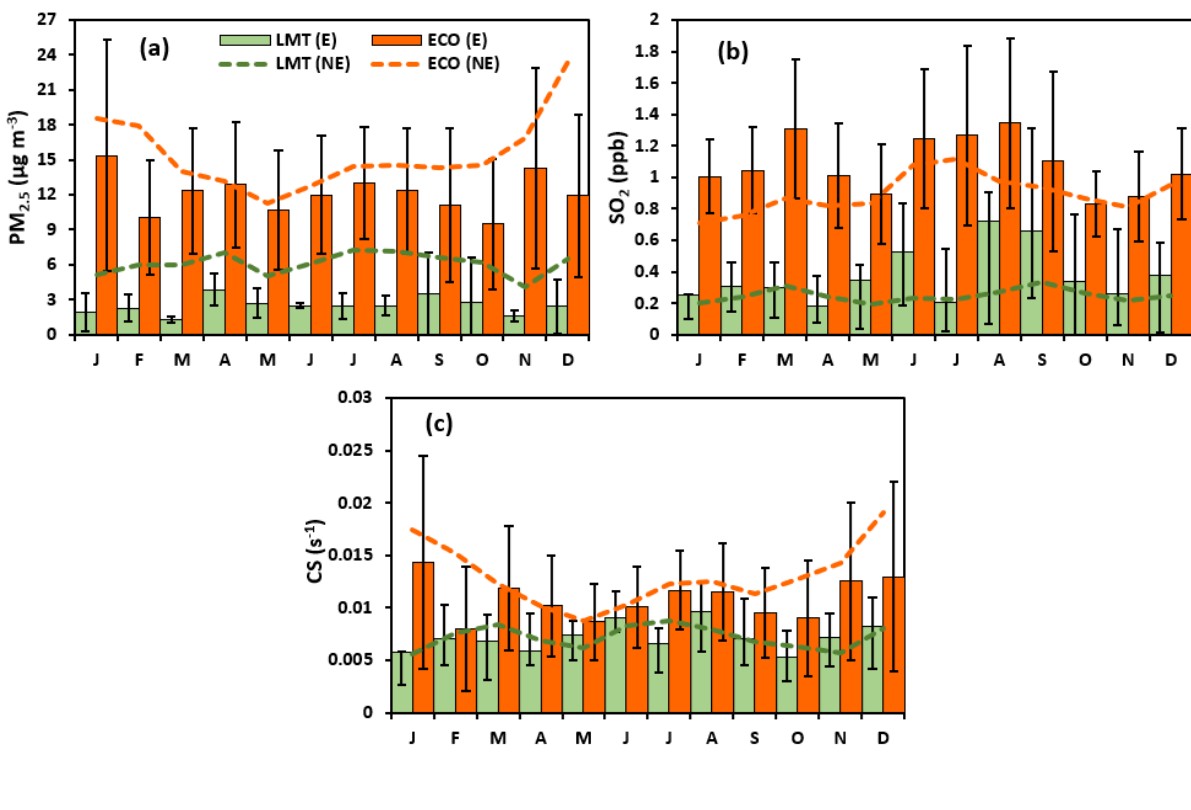

**Fig. 5. Monthly means of (a) PM₂.₅, (b) SO₂ and (c) CS related to event days (columns) and non-event days (dashed lines) for LMT (green) and ECO (red). The bars indicate the standard deviation. (d) Box plot of PM₂.₅, SO₂, and CS associated with nucleation days for LMT (green) and ECO (red). The box shows the quartiles, the median (the line inside the box), the mean (the star), and 90 th and 10 th percentiles (the whiskers).**





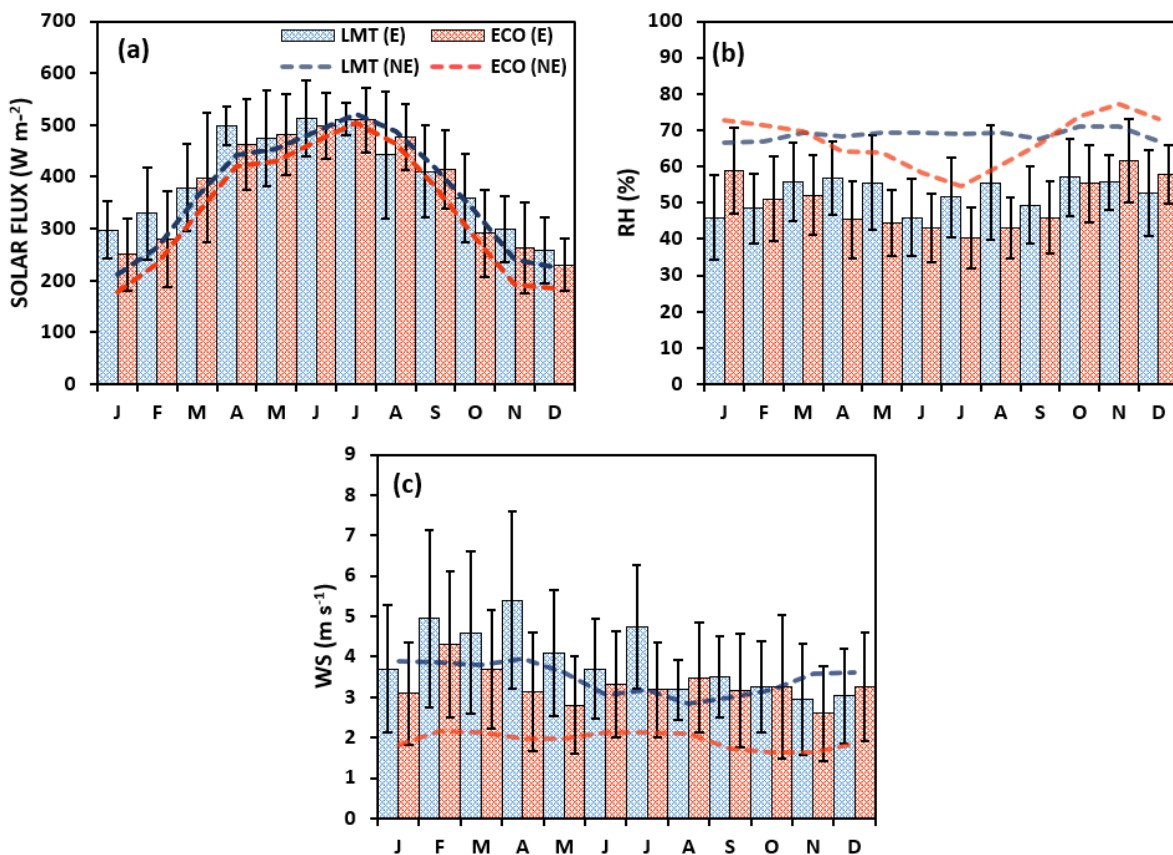


**Fig. 6. Monthly means of (a) solar flux, (b) relative humidity, and (c) wind speed related to event days (columns) and non-event days (dashed lines) for LMT (blue) and ECO (red). The bars indicate the standard deviation.**

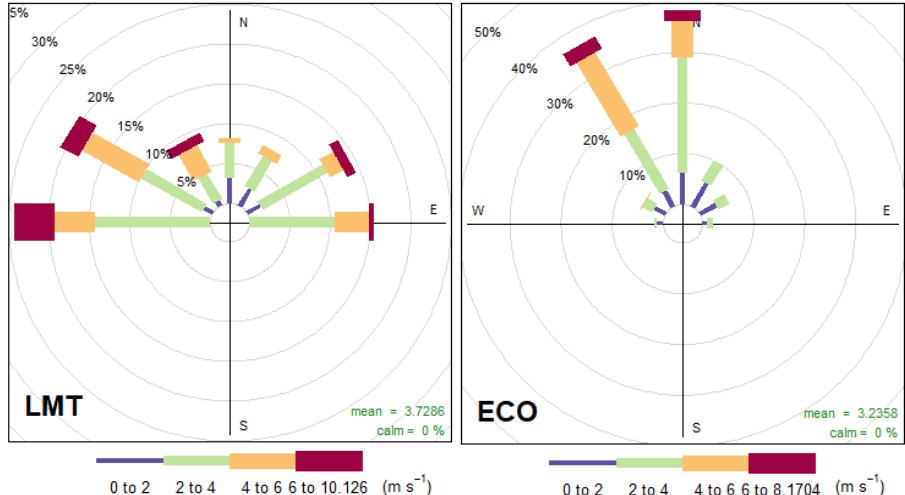






**Figure 7. Wind roses related to event days for LMT and ECO observatories.**

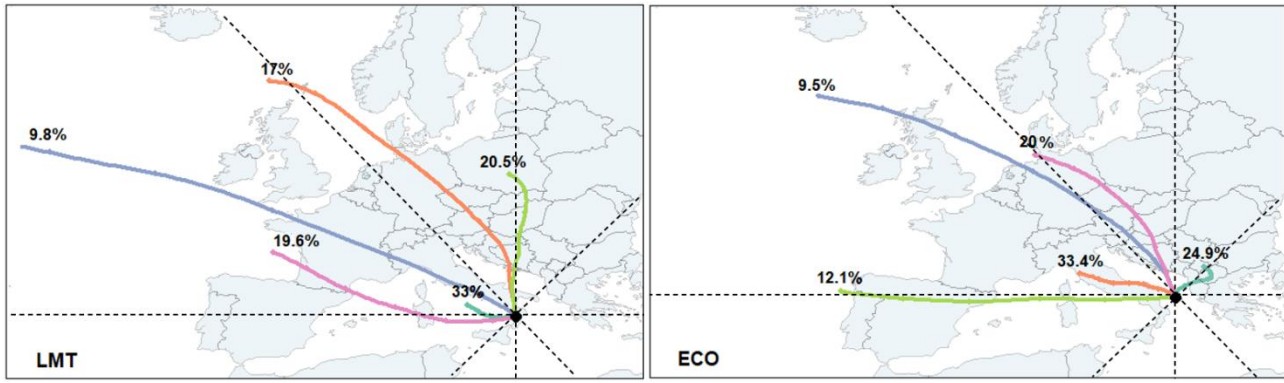


**Fig. 8. Cluster centroids retrieved from the 3-day analytical back trajectories reaching ECO and LMT at 500m above sea levels. The maps were plotted using Hysplit integrated with the Openair package.**




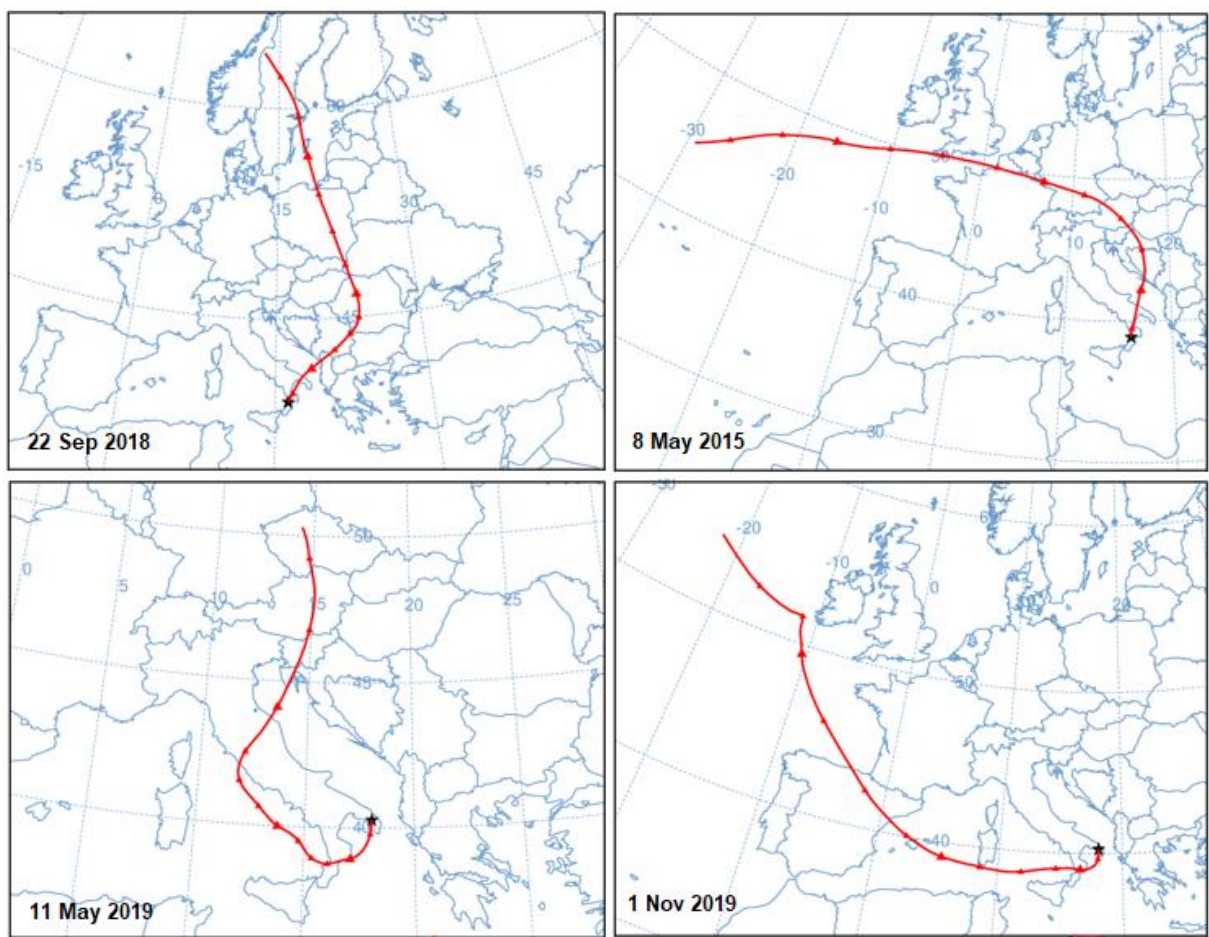

**Fig. 9. 72 h back-trajectories arriving at LMT (up) and ECO (down) during four simultaneous new-particle formation events. The maps were retrieved from NOAA Hysplit Model.**


**Tables**

**Table 1. Number of available days (N) and relative frequencies (f %) of each class, NPF events, non-events, and undefined events**
**detected in ECO and LMT observatories.**

|  | LMT (Total=1440) | | ECO (Total=1423) | |
| --- | --- | --- | --- | --- |
| Day Classification | Number | Frequency (%) | Number | Frequency (%) |
| NPF events | 124 | 9 | 352 | 25 |
| Non-events | 1272 | 88 | 1019 | 71 |
| Undefined events | 44 | 3 | 52 | 4 |





**Table 2. Arithmetic means ± 1 standard deviation and medians (25th–75th) percentiles of total, accumulation, Aitken, and nucleation PNC (cm$^{-3}$) for both sites.**

|  | Nnuc ($10^3$cm$^{-3}$) | Naitk ($10^3$cm$^{-3}$) | Nacc ($10^3$cm$^{-3}$) | Ntot ($10^3$cm$^{-3}$) |
|---|---|---|---|---|
| LMT | 1.1±1.0 | 2.2±1.1 | 1.2±0.6 | 4.4±2.2 |
|  | 0.7 (0.3-1.5) | 1.9 (1.4-2.7) | 1.0 (0.7-1.0) | 4.0 (2.8-5.6) |
| ECO | 1.6 ±1.0 | 4.1±1.8 | 2.1±1.4 | 7.8±3.4 |
|  | 1.3 (0.9-1.9) | 3.7 (2.8-4.8) | 1.8 (1.2-2.6) | 7.1 (5.5-9.3) |

**Table 3. Arithmetic means ± 1 standard deviation of nucleation, Aitken and accumulation PNC (cm$^{-3}$) related to event days (E) and non-event days (NE) for both sites.**

|  | LMT | | | ECO | | |
|---|---|---|---|---|---|---|
|  | Nnuc ($10^3$cm$^{-3}$) | Naitk ($10^3$cm$^{-3}$) | Nacc ($10^3$cm$^{-3}$) | Nuc ($10^3$cm$^{-3}$) | Naitk ($10^3$cm$^{-3}$) | Nacc ($10^3$cm$^{-3}$) |
| WE | 2.5±0.8 | 3.5±1.1 | 1.0±0.4 | 2.3±1.6 | 4.9±1.9 | 1.9±0.9 |
| WNE | 1.2±0.5 | 2.4±0.7 | 1.1±0.4 | 1.2±0.6 | 4.8±1.8 | 2.6±1.2 |
| SpE | 2.4±1.5 | 3.4±0.6 | 1.2±0.2 | 2.2±1.8 | 3.8±0.6 | 1.6±0.3 |
| SpNE | 0.8±0.4 | 2.0±0.4 | 1.2±0.2 | 1.1±0.3 | 3.0±0.7 | 1.6±0.3 |
| SE | 1.8±0.9 | 2.9±0.5 | 1.3±0.2 | 2.7±2.2 | 4.1±0.7 | 1.8±0.4 |
| SNE | 0.7±0.4 | 1.6±0.5 | 1.3±0.2 | 1.2±0.3 | 3.1±0.9 | 2.1±0.4 |
| AE | 2.0±0.9 | 3.3±1.2 | 1.0±0.5 | 2.4±1.2 | 5.2±2.2 | 1.9±1.1 |
| ANE | 1.1±0.4 | 2.3±0.6 | 1.1±0.3 | 1.5±0.6 | 4.9±1.9 | 2.7±1.1 |

**Table 4. Arithmetic means ± 1 standard deviation and medians (25 th–75 th) percentiles of different parameters, PM$_{2.5}$, SO $_2$, CS, GR, RH, WS and SF for both sites.**

|  | PM$_{2.5}$ (µg m$^{-3}$) | SO$_2$ (ppb) | CS (s$^{-1}$) x10$^2$ | GR (nm h$^{-1}$) | RH (%) | WS (m s$^{-1}$) | SF (W m$^{-2}$) |
|---|---|---|---|---|---|---|---|
| LMT | 3.1±2.6 | 0.3±0.3 | 0.6±3.0 | 6.7±2.6 | 48±10 | 4.2±1.4 | 394±110 |
|  | 2.0 (1.3-4.2) | 0.2 (0.2-0.4) | 0.6 (0.4-0.8) | 6.2 (5.0-8.2) | 50 (40-56) | 3.9 (3.0-5.0) | 360 (313-496) |
| ECO | 8.9±4.8 | 0.9±0.3 | 0.9±0.6 | 6.6±2.0 | 49±12 | 3.8±1.7 | 365±133 |
|  | 8.0 (5.1-11.0) | 0.9 (0.8-1.0) | 0.8 (0.8-1.0) | 6.6 (5.0-7.6) | 48 (41-56) | 3.5 (2.2-4.9) | 350 (340-470) |