# Peer review of "Characterization of ultrafine particles and the occurrence of new particle formation events in an urban and coastal site of the Mediterranean area"

_Atmospheric Chemistry and Physics, 2022_

## Author Comment (AC1)

Manuscript No. acp-2022-512

TITLE: "Characterization of ultrafine particles and the occurrence of new particle formation events in an urban and coastal site of the Mediterranean area".

Replies to Reviewer 2

The authors wish to thank the reviewer for his/her constructive suggestions and comments. We took them into careful consideration, and we hope that the revised version of the paper is improved. Each of the comments is addressed point-by-point below.

**Major comments**

1. **The manuscript is mainly descriptive, with some statistical analysis but no conclusions about the differences between both sites. I would suggest the authors to include some additional analysis to identify the factors that affect to the differences, specially on NPF events. The growth rate is not clear how it is retrieved or for which size range (see comment below), I would also suggest the authors to calculate the formation rate and include discussion about it. I expect to see some differences between both site on GR and formation rate, if there are differences, means that the vapours contributing to the formation and the growth are different at these sites. Also, I would suggest to include the analysis of H2SO4 instead of SO2 (solar radiation and CS are available and could estimate the sulfuric acid from proxies as Petäjä et al. 2009, ACP).**

   **REPLY: we followed the suggestions of the reviewer.**

   **The growth rate was explained better** "*Particle growth rate (GR) was calculated from time evolution of the mean geometric diameter $D_p$ in the size range of 10-20 nm, using Eq. (1) (Kulmala et al., 2012):*
   $$GR(nm\ h^{-1}) = (D_{p2} - D_{p1})/(t_2 - t_1) \tag{1}$$

   *with $Dp_1$ and $Dp_2$ the geometric diameter at the start time $t_1$ and end time $t_2$ of the growth event. Using the maximum concentration method, we identified the time when the concentration is at the maximum in each size bin. The growth rates were obtained as the slope of the linear fit of the times with the corresponding geometric mean diameters of the particles.*"

   **We also calculated the formation rate and included a discussion about it and GR** "*The growth rate (GR) and the particle formation rate (J) were analyzed to investigate the dynamic properties of NPF events. At the ECO site the growth rate values ranged from 3 nm h$^{-1}$ to 14 nm h$^{-1}$ (average 7.5 ± 3.3 nm h$^{-1}$) and J from 0.6 to 8.6 cm$^{-3}$ s$^{-1}$ (average 3.3 ±3.1 cm$^{-3}$ s$^{-1}$); while at LMT the GR varied from 2.5 to 10 nm h$^{-1}$ (average 6.1 ± 2.3 nm h$^{-1}$) and the J from 0.3 to 6.2 cm$^{-3}$ s$^{-1}$ (average 2.4 ± 1.8 cm$^{-3}$ s$^{-1}$). The values of both parameters are comparable with what was reported for NPF events in other urban and coastal sites (Hussein et al., 2020, Kalivitis et al., 2019; Salma et al 2019; Kalkavouras et al., 2020; Nieminen et al. 2018). In particular, similarities were found with some Mediterranean sites such as the coastal station of Finokalia GR~ 5 nmh$^{-1}$, J~ 0.9 cm$^{-3}$ s$^{-1}$ (Pikridas et al., 2012), the coastal/rural/suburban station of Akrotiri, GR~ 6 nmh$^{-1}$, J~13 cm$^{-3}$ s$^{-1}$ (Kopanakis et al., 2013) and the Cyprus Island, GR~ 2.8–5 nmh$^{-1}$, J~ 5–11.4 cm$^{-3}$ s$^{-1}$ (Debevec et al., 2018). The mean values of GR and J turned out to be higher at the ECO site than at the LMT site, and they showed a clear seasonal pattern with higher values during warm months, not observed at LMT where both parameters did not show distinctive features. As reported by Nieminen et al. (2018), the production rate of nucleation particles is generally higher in urban sites than in remote/clean ones because of the greater anthropogenic activity and the greater availability of precursors. In particular, the higher values of GR and J in warm months can be attributed to the intensification of the photochemical activity and abundance of SO$_2$ which acts as a precursor of sulfuric acid. The formation of nucleation mode particles is known to be influenced by the chemical and physical condition of the atmosphere*"

   **The analysis of H$_2$SO$_4$ was included and discussed** "*Sulphuric acid is identified as one of the key components directly connected to NPF process (Sipil̈a et al., 2010). Because no direct measurements of it were done in this study, we investigate its role, considering the proxy of sulphuric acid (Eq.4), without scaling factor (Petaja et al. 2009). The proxy only allows us to estimate the order of the average concentration levels of H$_2$SO$_4$ and although the results obtained are subject to uncertainties, they can still provide indications of trends (Salma et al., 2019). The average monthly values of H$_2$SO$_4$ proxy showed substantial differences between the two sites on event days (Fig 5d), 40 x10$^3$ ppbWm$^{-2}$s (ranging from 18 x10$^3$ to 61 x10$^3$ ppbWm$^{-2}$s) at ECO and 20x10$^3$ ppbWm$^{-2}$s (from 11 x10$^3$ to 38 x10$^3$ ppbWm$^{-2}$s) at LMT. These values are about 35% higher than non-events days in both cases. The proxy values of H$_2$SO$_4$ are larger in warm months and are substantially higher, by a factor of 2, in the urban*"

*background than in the coastal site, mainly due to the values of CS and SO₂. These last results are in accordance with GR and J values obtained for the two sites, where the low J and GR values at LMT could be associated with the low concentrations of H₂SO₄ (or other precursors). The conditions for the occurrence of NPF events are mainly driven by the ratio of the source and sink terms for the condensing vapors, therefore a greater availability of this gas precursor could have favored the occurrence of NPF events at ECO, although the higher values of CS, as well as the lower levels could have limited its development at LMT."*

2. **I recommend the authors to combine sections 3.4 and 3.5, and try to answer in these sections why there is that large differences in the event frequency in these two nearby locations? What promotes the regional NPF events? There is cases when NPF events are observed at both sites?**

   **REPLY: we combined sections 3.4 and 3.5 and tried to explain the differences found between the two sites**
   *"In general, these results underline the importance of specific atmospheric conditions (temperature, solar radiation, RH, origin of air mass, pollutant concentrations) under which the NPF events have occurred and emphasize how the observed differences are associated with the levels of pollution found in them. The more frequent NPF events at the urban background compared to the coastal site can be ascribed to a greater abundance of condensable species, deriving from anthropogenic emissions, which favor the growth of particles increasing their chance of survival. Regarding the different seasonality of the events, while the trend at the urban site of ECO can be associated with the increased photochemical activity and the higher concentrations of different precursors during the warm months, the seasonality at the coastal site of LMT is more difficult to explain. Along with the lower availability of precursors, local conditions could play an additional role as well. These may include synoptic systems such as increased turbulence during warm months and the different atmospheric composition (related to the proximity to the sea and the effects of land-sea breezes) due to which the newly formed particles could be more effectively suppressed preventing further growth."*

   **As already written in MS (lines 143)** *"Of all events detected during the study period, 50 were observed simultaneously at both sites".*
   **Also, a discussion about them was in section 3.4.** *"We also focused on "common" NPF events considering only the air masses pathway related to these days detected during the study period. From 3-days back-trajectory analysis, emerged that the back-trajectories of common events exhibit similar characteristics in terms of origin and pathways to what already observed, but in addition, two different cases were detected, in the first the trajectory passes from ECO before reaching LMT, while in the other case, vice versa the air mass passes from LMT and then reaches ECO. Out of 50 common NPF events, 31 were in the first case and 19 in the second. Two representative examples of back-trajectories for each case are depicted in Fig. 9 and show that there is not preferential path that can characterize them, because in both cases we find trajectories that come from both Eastern and Western Europe. These events occurred almost synchronously at the two sites, with a difference in starting time not greater than 30 minutes. The factors that characterized the concomitant events of NPF (Table 4) are similar to those observed for the non-concomitant events, with values of PM₂.₅, SO₂, CS, and H₂SO₄ proxy in LMT lower than ECO, and similar meteorological conditions. The simultaneous observation of these events indicates that the formation of new particles has a wide horizontal extension and can be seen as a large-scale phenomenon. It is probable that the air masses already contain particles that have been formed by nucleation somewhere and then transported. Or during their travel, the air masses are enriched with gaseous precursors deriving from anthropogenic emissions and/or from biogenic production such as to foster (potentially) NPF processes, even in locations far from the sources."*

3. **In the introduction the authors focus on the importance of regional NPF events, however, the manuscript lack of results and discussion on this topic. I would suggest to include more analysis on this topic, but if no further analysis, discussion, results are included about this, I would suggest to shorten that part in the introduction.**

   **REPLY:** as suggested we shortened part of the introduction focused on the importance of regional NPF events, deleting lines 47 to 53.

   **Minor comments**

   **L12 – change "occurred" by "occurring"?**
   REPLY: we changed "occurred" with "occurring".

**L89 – which different meteorological dynamics?**
REPLY: we changed "meteorological dynamics" with "local meteorological conditions". We referred mainly to the effect of the breezes, as written in the following sentence.

**L94-95 – please rewrite this sentence**
REPLY: we rewrote "*The LMT observatory is located far from the urban agglomeration and therefore is not directly affected by the emissions deriving from the main anthropogenic activities.*"

**L95-97 – would move this sentence after L89 about dynamics and would add other sentence about meteorological dynamic at ECO site.**
REPLY: we moved the sentence and rephrased as: "*Being on the coast, the local weather of LMT is influenced by a system of "land-see" breezes that guarantees a temperate climate and continuous ventilation throughout the year that favors an effective dilution of air pollutants.*"

**L103 – move this sentence with the next paragraph, where the authors present the quality control. Are the instrument routinely calibrated or psl checked or compared with total particle concentrations? Have the instruments been intercompared before?**
REPLY: we moved it. Both instruments are routinely calibrated for aerosol flow rate, sheath air flow rate, flow rate of the aerosol dryer's, leak test, DMA high-voltage check and psl check. The instruments have never been intercompared, but they were periodically subjected to the "round-robin test".

**L109 – multiple charged instead of negatively?**
REPLY: we corrected.

**Section 2 – I would recommend to name this section "Measurements and methods", then section 2.1 "Measurement sites and instrumentation" that unifies sections 2.1 and 2.2, and section 2.3 I would rename it as "Data analysis", "methodology", "methods",… "Evaluation of NPF events", I think is not the most appropriate. Include in this section the formulas for the CS that is later discussed.**
REPLY: The suggestions were adopted.

**L132 – the authors use frequently paragraphs of just one sentence, please avoid this.???**
REPLY: thanks for the advice.

**Table 1 includes Events, Undefined and Non-event days, that sum the total number of days. However, line 134 says that there is a 78% of data coverage? How can classify more days than the data coverage (~0.78\*TotalNumberDays).**
REPLY: we apologize for the misunderstanding. We mean that the study is based on a period of 5 years, 1826 total days. Since the measurement we collected were 1423 days at ECO and 1440 at LMT, the coverage over 1826 days is approximately 78%. We made it explicit in the MS "*Over five years of measurements, we had a data coverage of ~78 %, where the available measurement days were 1423 at ECO and 1440 at LMT.*"

**L140 – "confirming what was already observed in Dinoi et al. (2021a)" I would suggest to rephrase, something like "showing similar results than those presented by a shorter measurement period presented by Dinoi et al. (2021a)".**
REPLY: as suggested we re-phrased "confirming what was already observed in Dinoi et al. (2021a)" with "showing similar results to those found in a shorter measurement period presented by Dinoi et al. (2021a)".

**L146 – where these numbers come from?**
**REPLY:** they come from various studies carried out in the Mediterranean area. We rephrased *"The annual frequencies of NPF (9% and 25 %) are in good agreement with frequencies (10 % - 36 %) found in other studies, based on long-term measurements, carried out in the Mediterranean area (Kopanakis et al., 2013; Kalivitis et al., 2019; Hussein et al., 2020; Kalkavouras et al., 2020; Baalbaki et al., 2021)."*

**L156-160 – GR is a quantity that depends on the diameter. Here the authors don't define the diameter range where the GR is being retrieved. If the GR changes with time, probably because the diameter range change?**
REPLY: we defined in the MS the diameter range (10nm-20nm) where the GR is being retrieved.

**L162 – avoid the term "emission levels", mainly because the authors are not really measuring emissions, only measuring atmospheric concentrations.**
REPLY: we deleted "emission levels" and replaced it with "concentrations".

**L166 – use the correct significant numbers, the table is correct. Same in the following paragraphs.**
REPLY: thanks for the suggestion

**L176 – add space before ~3100**
REPLY: we did it

**Eq2 – use subindex for E and NE.**
REPLY: we did it

**L213-218 – I would add some references were this method has been previously used at different locations and compare how important NPFs are in other locations compared to those presented in this work (e.g., Bousiotis et al., 2021; Casquero-Vera et al., 2021; Thén et al., 2022). Are the values reported averages for the NPF time of for the whole day?**
REPLY: we added the suggested references and compared the results *"From the coastal to urban background site, we found a decrease in the contribution of NPF events to particle number, similar to what was observed by Salma et al., (2017) between the near city background (2.3) and the city center (1.6) of Budapest over 5 years. In the study of Bousiotis et al., (2021), on 13 sites from five countries in Europe it was found that for almost all rural background sites $NFS_{NUC}$ was greater than 2, and reached 4 in a very clean site of Finland. Nemet et al (2018) found lower values of $NFS_{NUC}$, 1.58, 1.54, and 2.01, in the cities of Budapest, Vienna and Prague, respectively, while in Granada urban site $NFS_{NUC}$ was 1.05 (Casquero-Vera et al., (2021). The decrease in the contribution of NPF events to particle number, moving from a more polluted to a less polluted site, may be related to the higher contribution to particle number concentrations of other sources, i.e. traffic and heating, and the associated increased condensation sink."*

The values reported are averages for the whole day.

**L240 – I would not say is surprising, if there is less CS, probably there is also less precursor vapors too…**
REPLY: the reviewer is right. We removed the sentence and rephrased *"Therefore a greater availability of this gas precursor ($H_2SO_4$) could have favored the occurrence of NPF events at ECO, although the higher values of CS, as well as the lower levels could have limited its development at LMT."*

---

## Author Comment (AC2)

Manuscript No.   acp-2022-512

TITLE:    "Characterization of ultrafine particles and the occurrence of new particle formation events in an urban and coastal site of the Mediterranean area"

Replies to Reviewer 1

The authors wish to thank the reviewer for his/her constructive suggestions and comments. We took them into careful consideration, and we hope that the revised version of the paper is improved. Each of the comments is addressed point -by-point below.

**RC1: 'Comment on acp-2022-512', Anonymous Referee #1, 27 Sep 2022  reply**
**The MS deals with the properties of NPF events and particle number concentrations in various size fractions at an urban background site in Lecce and at a costal location in central Mediterranean, southern Italy. It presents valuable results and conclusions and contributes to the growing knowledge on the atmospheric nucleation and consecutive particle growth phenomenon in this larger region. However, the MS could be and should be improved substantially in several ways before deciding whether it is acceptable or not for publication in the ACP. The corrections can hopefully be accomplished by a very careful and thorough revision of the present version.**

**Major concerns**

1. **The MS is too much of the descriptive character. Lots of simple statistical results are just supplied without interpreting them or putting them into appropriate frameworks or formulating clear conclusions or messages from them. Examples could be large parts of Sect. 3.2, lines 221–228 or Table 3. Further possibilities for improved interpretations could involve e.g. explaining and better comparing the seasonality of NPF events and diurnal concentration patterns of various particle number size fractions. Further important references on urban NPF could also be added to this purpose.**
   **REPLY:** as suggested, the manuscript was shortened in some places, including from lines 221 to 225. Also, we added the following references: Putaud et al., 2010; Asmi et al., 2011; Kalivitis et al., 2019; Casquero-Vera et al., 2020; Kalkavouras et al., 2020. Additional comments were added to better explain the seasonality of the events and the contribution of NPF events on particle number concentrations.
   **Sect. 3.2 was modified:** *"A clear diurnal pattern in each mode particle number concentration was observed in every season. Fig. 4 shows the trend of each mode fraction considering separately the days of NPF events ( E, solid line) and the days of non-events (NE, dashed line). The timing of measurements is expressed in solar time (UTC + 1). Nucleation, Aitken, and accumulation mode particles have very similar behavior during non-events and, except for the different concentrations, both sites show a pronounced diurnal cycle with a morning and evening peak. The two peaks are shifted by one hour between spring-summer and autumn-winter because of daylight savings time and are mainly linked to vehicular emissions, most intense during the morning and evening rush hour. In addition, the evening peaks of Aitken and accumulation mode particles in winter and autumn can be linked to domestic heating emissions, mainly biomass burning considered an important source of ultrafine particles in urban sites. In cold months the pollutants tend to accumulate during the night due to the reduced boundary layer compared to the daytime layer. These peaks are also present in LMT but are less intense due to the greater distance of the site from the urban centre. Regarding event days, in both sites together with the two peaks of rush hours, nucleation mode particles present further picks around noon, more marked in summer, spring, and winter in ECO, and spring, summer, and autumn in LMT, and less marked instead in winter and autumn in LMT and ECO respectively. Similar observations have been reported in Cusack et al. (2013), Kalivitis et al. (2019), Kalkavouras et al. 2020, Dinoi et al. (2020, 2021a), for the western Mediterranean sites where the diurnal variation in nucleation mode particles presents a clear maximum at noon under both polluted and clean conditions. The contribution of the NPF process to the number concentration is also observed in the Aitken mode particles, more noticeable in the LMT site with 30 % in autumn-winter and 41 % in spring-summer, and with 21 % only in spring-summer in the ECO site. Nucleation mode particles show an increase of 52 %, 65 %, 61 %, and 49 % in winter, spring, summer, and autumn in LMT, and of 47 %, 52 %, 55 %, and 39 % in ECO. These results highlight that the formation of new particles contributes to the overall particle population more in the warm months and more significantly in the coastal site than in the urban background site, probably because the urban site is also affected by local emission of ultrafine particles that tend to suppress the NPF process. No contribution is observed in the concentration of accumulation mode particles where especially in the first half of the day, the concentrations were higher on non-event than event days, especially at the ECO site. This could explain the*

*different frequency of events that characterized the two sites in these seasons, assuming that the NPF events were favored on those days with lower particle number concentrations (Salma et al., 2017)."*

2. **The $SO_2$ is often used in the existing interpretations (e.g. in Sect. 3.3). Despite the fact that 1) its photochemical oxidation to the key nucleating vapour of $H_2SO_4$ was shown to be slow, complex and of less direct influence on the NPF, and 2) the authors possess all necessary properties and variables for deriving the proximity value of $H_2SO_4$ (which is more directly connected to the process) either by the classical method of Petäjä et al. (ACP, 2009) without the scaling factor or by its improved estimation proposed in Dada et al. (ACP, 2020). The authors may want to amend this part, which could contribute to the improved overall quality of the final MS.**
   **REPLY: as suggested, we included the analysis of $H_2SO_4$ and discussed it.**
   **In 2.2 Data analysis we added** *"Sulfuric acid ($H_2SO_4$) is considered a key precursor for new particle formation, therefore its concentrations were derived by calculating the $H_2SO_4$ proxy ($ppbWm^{-2}s^{-1}$), without scaling factor, using the method presented by Petäjä et al. (2009):*

$$[H_2 SO_4]\alpha \frac{SO_4 \; x \; SRad}{CS} \tag{4}$$

*where SO2 is the sulphur dioxide concentration, SRad is the solar radiation flux, and CS the condensation sink.*

   **In Sect. 3.3 we added** "*Sulphuric acid is identified as one of the key components directly connected to NPF process (Sipilä et al., 2010). Because no direct measurements of it were done in this study, we investigate its role, considering the proxy of sulphuric acid (Eq.4), without a scaling factor (Petaja et al. 2009). The proxy only allows us to estimate the order of the average concentration levels of $H_2SO_4$ and although the results obtained are subject to uncertainties, they can still provide indications of trends (Salma et al., 2019). The average monthly values of $H_2SO_4$ proxy showed substantial differences between the two sites on event days (Fig 5d), $40 \; x10^3$ $ppbWm^{-2}s$ (ranging from $18 \; x10^3$ to $61 \; x10^3$ $ppbWm^{-2}s$) at ECO and $20x10^3$ $ppbWm^{-2}s$ (from $11 \; x10^3$ to $38 \; x10^3$ $ppbWm^{-2}s$) at LMT. These values are about 35% higher than non-events days in both sites. The proxy values of $H_2SO_4$ are larger in warm months and are substantially higher, by a factor of 2, in the urban background than in the coastal site, mainly due to the values of CS and $SO_2$. The conditions for the occurrence of NPF events are mainly driven by the ratio of the source and sink terms for the condensing vapors, therefore a greater availability of this gas precursor could have favored the occurrence of NPF events at ECO, although the higher values of CS, as well as the lower levels, could have limited its development at LMT.*"

3. **It was shown in several publications that the size range below 10 nm is crucial for identifying and characterizing NPF events in particular in cities (e.g. Nieminen et al., ACP, 2018). The authors are asked to discuss how they avoided the limitations imposed by their relatively large measurable diameter of 10 nm in Lecce. For instance, how did this fact influence the share of the undefined days?**
   **REPLY:** The reviewer is right, the size range below 10 nm is very important for studying the beginning of NPF and characterizing the early stages of growth. Therefore, in this case, the events in which new particles were unable to grow beyond 10 nm could not be identified. However, as widely reported in many studies in the literature, a detection limit of 10 nm in diameter does not prevent the correct identification and characterization of those events whose growth stage of newly formed particles fully falls in the studied size range.

4. **The frequency of missing days was relatively large around 22% at both sites. It is wondered how these days were distributed over the years or the measurement campaign since the NPF occurrence frequency showed a remarkable seasonal dependency, which could possibly impact the representativity of the remaining days.**
   **REPLY:** as suggested by the reviewer, we checked how the missing days were distributed over the years. The following figures show that the lack of data is mainly in the months of February/March for LMT, and January for ECO. Therefore, we believe that the data are representative on yearly and monthly timescales for all months except those just mentioned which, due to the larger ratio of missing days, could be less representative.

[Figure]

[Figure]

5. **The reader can have the feeling at several places (e.g., in lines 154–160) that the nucleation or NPF processes and the particle growth process are not clearly distinguished. For instance, it could be clarified what the authors meant by "the temporal evolution of the events".**
   **REPLY:** during the NPF process, a marked increase in the number concentration of particles in the nucleating mode is observed, followed by their growth. So, with "temporal evolution of the events" we mean the temporal evolution of the particle number size distribution and the respective variation of the geometric mean diameter ($D_p$) of nucleation mode particles.

6. **It is not described properly how some important properties were obtained. An example could be the lines 222–224 where the CS is mentioned only very briefly. By the way, this (in a more detailed extent) should be shifted from the section Results and discussion to e.g. Sect. 2.3 since this is not their result. In this respect, it is also mentioned that the original NSF in Salma et al., ACP, 2017 was further developed, and its NSF$_{GEN}$ and NSF$_{NUCL}$ are more informative than the original form and should be used or at least mentioned. It is not clear (lines 225–228) how the start and end times of the growth events and more importantly, the geometric diameters $D_{p1}$ and $D_{p2}$ were derived and whether the latter were modal median diameters or something else.**
   **REPLY: the CS was detailed and moved to Section 2.3.**

   *"The condensation sink, CS ($s^{-1}$), quantifies how rapidly a condensable gaseous compound condenses on available aerosol particles (Kerminen et al., 2018), and then it depends on the effective surface area of pre-existing particles. CS was calculated, using the methods available in the literature (Dal Maso et al., 2005, and references therein) considering sulphuric acid ($H_2SO_4$) as the condensable species:*

   $$CS = 2\pi D \sum_{D'_p} \beta_m (D'_{pi}) D'_{pi} N_i \qquad (2)$$

   *where D is the diffusion coefficient for $H_2SO_4$, $N_i$ is the particle number concentration with diameter $D'_{pi}$ of the size bin i, and $\beta_m$ is the transition correction factor (Fuchs et al., 1971)."*

   **As suggested, we mentioned NSF$_{GEN}$ and integrated more information** *"The relative increase in particle number concentration due to the NPF process was also quantified with the nucleation strength factor (NFS) proposed by Salma et al. (2016, 2017, 2019). It measures the effects of nucleation events on ultrafine particles at a site considering two factors, NSF$_{NUC}$ that provides a measure of the concentration increment on nucleation days exclusively caused by NPF, and NSF$_{GEN}$ gives a measure of the overall contribution of NPF over a longer time span. In this work, we considered only NSF$_{NUC}$, calculated following Eq. (5) ..."*
   *"From the coastal to urban background site, we found a decrease in the contribution of NPF events to particle number, similar to what was observed by Salma et al., (2017) between the near city background (2.3) and the city center (1.6) of Budapest over 5 years. In the study of Bousiotis et al., (2021), on 13 sites from five countries in Europe it was found that for almost all rural background sites NFS$_{NUC}$ was greater than 2, and reached 4 in a very clean site of Finland. Nemet et al (2018) found lower values of NFS$_{NUC}$, 1.58, 1.54, and 2.01, in the cities of Budapest, Vienna and Prague, respectively, while in Granada urban site NFS$_{NUC}$ was 1.05 (Casquero-Vera et al., (2021). The decrease in the contribution of NPF events to particle number, moving from a more polluted to a less polluted site, may be related to the higher contribution to particle number concentrations of other sources, i.e. traffic and heating, and the associated increased condensation sink."*

   **Related to growth events we integrated** *"Particle growth rate (GR) was calculated from time evolution of the mean geometric diameter $D_p$ in the size range of 10-20 nm, using Eq. (1) (Kulmala et al., 2012):*
   $$GR(nm\ h^{-1}) = (D_{p2} - D_{p1})/(t_2 - t_1) \qquad (1)$$

   *with $D_{p1}$ and $D_{p2}$ the geometric diameter at the start time $t_1$ and end time $t_2$ of the growth event. Using the maximum concentration method, we identified the time when the concentration is at the maximum in each size bin. The growth rates were obtained as the slope of the linear fit of the times with the corresponding geometric mean diameters of the particles."*

7. **Figure 4 (and possibly some others as well) contains too many lines and it is difficult to follow. In addition, it should be discussed whether a local time involving the daylight-saving time (clock change in the EU) or UTC+1 or else were used as the time scale. This could be related to the shift in the positions of the diurnal peaks in different seasons.**
   **REPLY:** we split Figure 4 into multiple single plots.

[Figure]

**Fig. 4. Average diurnal variation in nucleation, Aitken, and accumulation mode particle number concentration (from top to bottom) over the whole period of study at LMT and ECO. Solid lines are for NPF events days (E) and dashed lines for non-events days of (NE), red for winter, grey for spring, blue for summer, and yellow for autumn.**

In both observatories, all the measurements were carried out maintaining the solar time throughout the years, therefore UTC + 1. We specified in the text that *"The timing of measurements is expressed in solar time (UTC + 1)."*
Also that *"The two peaks are shifted by one hour between spring-summer and autumn-winter because of daylight savings time"*

8. **Line 239 and further: the phenomenon or process is somewhat more sophisticated. The authors perhaps want to include and discuss the NPF occurrence with respect to the ratio of sources and sinks of low-volatility vapours and not just the amount of CS alone.**
   **REPLY:** line 239 to 250 were removed. We included the comment *"The conditions for the occurrence of NPF events are mainly driven by the ratio of the source and sink terms for the condensing vapors, therefore a greater availability of this gas precursor could have favored the occurrence of NPF events at ECO, although the higher values of CS, as well as the lower levels, could have limited its development at LMT."*

**Minor comments**

9. **Line 33: use either primary or emission (source).**
   **REPLY:** we deleted "emission"

10. **The references should be ordered chronologically and not alphabetically, e.g. lines 36–37.**
    **REPLY:** thanks, we ordered the references.

11. **Some abbreviations are not explained, e.g. line 101: MPSS, or Table 3 W, Sp, S and A.**
    **REPLY:** the abbreviation MPSS (Mobility Particle Size Spectrometer) was explained in the abstract but now we also added in line 101. At the same way, the abbreviation (winter W, spring Sp, summer S, and autumn A) was explained in the line 200 and now we also added in caption of Table 3.

12. **Line 105: is TSI Inc. really based in Rome, Italy?**
    **REPLY**: the reviewer is right, it is based in USA.

13. **Line 239: replace "discouraged" by not favoured or something similar.**
    **REPLY:** we replaced "discouraged" by not favoured.

**14. What is the advantage of using a CNR4 net radiometer which measures the energy balance between incoming short-wave and long-wave far infrared radiation versus surface-rejected short-wave and outgoing long-wave radiation to measuring global or direct solar radiations.**

**REPLY:** The CNR4 consists of two pyranometers which measure the solar radiation both incoming and reflected, and two pyrgeometers which measure the far Infrared radiation. From a spectral point of view, the pyranometer and pyrgeometer are complementary and together cover the full spectral range, the pyranometer from 0.3 to 3 microns, and the pyrgeometer from 4.5 to 42 microns. The advantage of using this kind of instrument is that all components are measured separately, and this allows every single parameter to be used according to specific work requirements.

---

## Author Response (AR2)

Manuscript No.   acp-2022-512

TITLE:   "Characterization of ultrafine particles and the occurrence of new particle formation events in an urban and coastal site of the Mediterranean area"

Replies to Reviewer 2

The authors wish to thank both referees for their constructive comments and good suggestions to improve our MS. Below answers to referee comments and changes made (yellow in the revised text) for the MS.

Minor comments

L95 – As it is written seems is the same instrument, maybe change for "The instrumentation used are the same instrument model...".
**REPLY: we changed it.**

L127 – change by "(Fuchs and Sutugin, 1971)"? Same in other parts of the manuscript for Lehtinen et al. 2003, "Lehtinen and Kulmala, 2003"?
**REPLY: we changed them.**

L128 – I would add "J ($J\_10$; cm$-3$s$-1$)" just to clarify, it is the common terminology.
**REPLY: we added it.**

L184 – A clear diurnal pattern "for"?
**REPLY: we corrected it.**

L268 – Sipilä et al., 2010
**REPLY: we corrected it.**

L269 – Petäjä et al., 2009; "Eq. 4" the space
**REPLY: we corrected it.**

L272 – "Fig. 5d" the dot
**REPLY: we corrected it.**